# Decellularization of Full Heart—Optimizing the Classical Sodium-Dodecyl-Sulfate-Based Decellularization Protocol

**DOI:** 10.3390/bioengineering9040147

**Published:** 2022-04-01

**Authors:** Reem Al-Hejailan, Tobias Weigel, Sebastian Schürlein, Constantin Berger, Futwan Al-Mohanna, Jan Hansmann

**Affiliations:** 1Department of Cell Biology, King Faisal Specialist Hospital and Research Centre, Riyadh 11211, Saudi Arabia; rhijailan@kfshrc.edu.sa (R.A.-H.); futwan@kfshrc.edu.sa (F.A.-M.); 2Department of Tissue Engineering and Regenerative Medicine, University Hospital Würzburg, 97070 Würzburg, Germany; tobias.weigel@uni-wuerzburg.de (T.W.); sebastian.schuerlein@gmail.com (S.S.); constantin.berger@uni-wuerzburg.de (C.B.)

**Keywords:** tissue engineering, decellularization, vascularized scaffold, cardiac patch, dynamic culture

## Abstract

Compared to cell therapy, where cells are injected into a defect region, the treatment of heart infarction with cells seeded in a vascularized scaffold bears advantages, such as an immediate nutrient supply or a controllable and persistent localization of cells. For this purpose, decellularized native tissues are a preferable choice as they provide an in vivo-like microenvironment. However, the quality of such scaffolds strongly depends on the decellularization process. Therefore, two protocols based on sodium dodecyl sulfate or sodium deoxycholate were tailored and optimized for the decellularization of a porcine heart. The obtained scaffolds were tested for their applicability to generate vascularized cardiac patches. Decellularization with sodium dodecyl sulfate was found to be more suitable and resulted in scaffolds with a low amount of DNA, a highly preserved extracellular matrix composition, and structure shown by GAG quantification and immunohistochemistry. After seeding human endothelial cells into the vasculature, a coagulation assay demonstrated the functionality of the endothelial cells to minimize the clotting of blood. Human-induced pluripotent-stem-cell-derived cardiomyocytes in co-culture with fibroblasts and mesenchymal stem cells transferred the scaffold into a vascularized cardiac patch spontaneously contracting with a frequency of 25.61 ± 5.99 beats/min for over 16 weeks. The customized decellularization protocol based on sodium dodecyl sulfate renders a step towards a preclinical evaluation of the scaffolds.

## 1. Introduction

Despite remarkable progress in cell, tissue, and organ transplantation in the past decades, many challenges in regenerative medicine still remain, e.g., (i) limited long-term biocompatibility and biofunctionality of implants, (ii) shortage of organ donation and (iii) immune rejection of the engraft. Several clinical trials in myocardial infarction treatment that involved stem cells showed promising results, however, cell survival and homing to the proper environment are often insufficient. As a consequence, most of the cells die within the first week after the injection of cell suspension [1]. Delivering cells in a sheet rather than in a suspension resolved the issue to a certain extent by improving cardiac function and reducing fibrosis [2]. However, matrices of collagen, fibronectin, and matrigel have not achieved sufficient long-term success due to a lack of biochemical cues compared to the native environment [3]. Moreover, vascular structures for nutrient supply are absent in simple matrices. 

Natural biological scaffolds are more appropriate for this purpose as they provide an in vivo-like microenvironment supporting cell attachment, proliferation, and maturation [4]. In the recent decade, various extracellular matrix (ECM) scaffolds were generated by the decellularization of native tissues. Since porcine organs present a suitable donor material for human xenotransplantation [5], porcine tissues are also a source for matrices that emulate an organ-specific structure and biochemical microenvironment. For decellularization, native tissues are exposed to chemical and enzymatic agents as well as physical stimulation, such as sonication, freezing, and thawing with agitation to disrupt cell membranes and facilitate the removal of cell remnants from the ECM [6]. For example, a protocol utilizing Triton X-100, sodium dodecyl sulfate (SDS), sodium deoxycholate, CHAPS, and Tween 20, over a range of concentrations, was successfully used for porcine aortic valve leaflet decellularization [7]. The decellularization of whole porcine hearts requires a more advanced process to ensure complete cell removal from various heart compartments. This can be achieved by retrograde coronary perfusion in combination with a series of enzymes, detergents, and acids. Retrograde perfusion could also be complemented by hypertonic or hypotonic rinses, along with high flow rates to facilitate cell lysis and removal with effective tissue clearance [8,9]. However, protocols for the decellularization of whole hearts so far involved several chemicals perfusions and mechanical or physical agitation applications. This can nullify matrix proteins and their biochemical cues; otherwise, it is further time- and resource-consuming [10,11]. Thus, choosing the appropriate decellularization agents and reducing the exposure time to improve ECM preservation is crucial to generating highly bioactive heart scaffolds. 

Although many studies were successful in generating cardiac tissues using biological scaffolds generated from decellularized hearts and human-induced pluripotent stem cells (hiPSc) [12,13], none of them supported long-term survival, due to poor nutrient and oxygen supply. Since the main objective is to treat ischemic cardiac areas, it is necessary to ensure adequate oxygen and nutrient supply to the graft following implantation to prevent attrition. Thus, recent research strived for establishing 3D vascularized cardiac tissues. For example, microvascular endothelial cells (mVEC), human mesenchymal stem cells (hMSC), and hiPSc-cardiac cells (hiPSc-CC) were combined on a collagen cell carrier. This 3D cardiac patch demonstrated neo-angiogenesis and neo-cardiogenesis in vitro [14]. Moreso, 3D-printed scaffolds using a bioink called hdECM supplemented with 10 µg/mL of VEGF and human hiPSc were generated, leading to enhanced cardiac repair, cardiac progenitor cells (CPC) maturation, and migration to the infarcted area for four weeks. Although the bioink dECM supports cellular function, their bounded shape fidelity, and weak mechanical integrity hinder the construction of large muscular tissue. To overcome the drawbacks of these approaches, a decellularized biological scaffold, including a vascular structure, could be an approach that enables the integration and engrafting by host tissue.

In order to bear a suitable microenvironment, and ensure nutrients and oxygen supply through a functional vasculature, our previous work [15] focused on the development of the first cardiac patch that supports perfusion with blood using porcine decellularized BioVasc, a derivative of small intestinal submucosa (SIS). This matrix provides a vasculature that can be anastomosed to a host’s circulatory system and serve as a scaffold to co-culture (hMSC), fibroblasts, and hiPSc-CC with clinical potential. The patch showed robust functionality in culture for four months. It is, however, becoming apparent that differences in ECM density, orientation, and structure, as well as composition, indicate diverse characteristics of the decellularized small intestine and decellularized cardiac tissue in terms of biochemical, structural, and mechanical properties. For example, a full-thickness cardiac tissue based on SIS is impossible due to the significant thinner wall thickness of the intestine compared to cardiac muscle. Thus, the scaffold which is based on porcine intestine contradicts the objective to provide a tissue-specific scaffold structure and an in vivo-like microenvironment.

In summary, previous approaches for generating vascularized cardiac patches, using a tissue-unspecific scaffold, did not reflect the macro- and micro-environmental cues of the original tissue. The adequate generation of decellularized heart tissue presents another obstacle and was mostly achieved by the retrograde perfusion through the aorta using alternating cycles of various chemical and enzymatic agents, including SDS, SDO, Triton X-100, and trypsin thus far [16]. Those approaches resulted in insufficient removal of xenoantigenic epitopes and/or a massive loss of ECM components [17]. The aim of this study was to identify an optimal decellularization protocol that requires less chemical treatment and process steps, facilitates the generation of a vascularized scaffold with cardiac-specific structures and microenvironment, enables long-term survival of a graft, and thereby paves the way for clinical use.

## 2. Materials and Methods

### 2.1. Tissue Explantation and Decellularization

Decellularization of the heart: The decellularization protocol was adapted from Ott et al. [18] and slightly modified. All animals were treated in compliance with care and the use of the Laboratory Animals Guide and the approval of the Institutional Board of Animal Protection and the German Society for Animal Care. The piglets (3 months old, 15–25 kg) were anesthetized using trapanal fentanyl, heparinized (1000 UE/Kg), and sacrificed using T61^®^. A median sternomy cut was made to expose the pericardium and ligate the heart through the ascending aorta using a reducing connector to fit around the aorta. The heart was then placed in a 5-L chamber and retrograde coronary perfusion was started with heparinized PBS^−^ under the constant pressure of 60 mmHg for 4–5 h. Following, two protocols were applied for decellularization:

**Protocol** **1.**
*Where 4% SDO (Sigma-Aldrich, Munich, Germany) in MilliQ water was applied for seven days of decellularization, followed by 3 h of MilliQ water and one day with 1% Triton X-100 in MilliQ water afterward. Final rinsing with PBS^−^/antibiotics solution was performed until day 14.*


**Protocol** **2.**
*Where 1 % SDS (Carl Roth GmbH, Karlsruhe, Germany) in MilliQ water was used to decellularize the heart for 3 days, followed by 3 h of MilliQ water rinsing and one day of 1% Triton X-100 (Sigma-Aldrich, Munich, Germany) in MilliQ water. On day 5, PBS^−^/antibiotics solution was started until day 14.*


In both protocols, gamma sterilization was conducted 14 days after organ explantation at a dosage of >25 kG using the sterilization service provided by BBF steriXpert (Kernen-Rommelshausen, Germany) and sterilized samples were stored in PBS^−^ at 4 °C.

### 2.2. DNA Quantification and Qualitative Assessment of Tissue Clearance

dsDNA was extracted from weighed wet tissue pieces using the DNeasy^®^ Blood and Tissue Extraction kit (Qiagen, Hilden, Germany) according to the manufacturer’s guidelines. Sample elution was carried out with a volume of 100 µL. DNA content was quantified using a NanoQuant Plate™ in combination with an Infinite^®^ 200 PRO plate reader (TECAN, Männedorf, Switzerland). DNA from native tissue was measured at a 1:5 dilution (*n* = 3). 

### 2.3. Quantification of Glycosamine Glycans (GAG) Content

For further quantification of extracellular matrix, a total glycosamine glycans (GAGs) assessment was carried out from lyophilized normal and decellularized tissue using the colorimetric Blyscan sGAG kit according to the manufacturer’s instructions (Biocolor, UK) (*n* = 3).

### 2.4. Comparison of Mechanical Properties of Porcine Decellularized Matrix from the Heart and the Small Intestine

The process of small intestine decellularization was published previously [15]. These scaffolds served as control. Scaffold samples were cut into strips of at least 40 mm in length and approximately 8 mm in width. Stress–strain measurements were performed in a Zwick universal testing machine (Zwick Roell, Ulm, Germany/10 kN load cell), with ambient temperature as well as humidity and a clamping length of 30 mm. The samples were clamped in the testing machine directly from the storing solution (PBS) without any drying treatment. As adhesive support, a rough abrasive paper was fixed in the clamping unit. Before the measurement, the size of the cross-section was determined by a caliper square. The stress–strain measurement was performed with a separation speed of 10 mm/min until the failure of the tissue sample. The measured force was converted into mechanical stress using the estimated cross-section before the testing. As Young’s modulus is defined by the slope of the linear region, two values were applied from the linear region of the stress–strain curve. The stress–strain diagram showed two linear areas. Therefore, two different Young’s moduli were determined and assigned to the gelatinous and fibrous tissue state, respectively. Five stress–strain diagrams of each tissue were analyzed for the evaluation of Young’s modulus.

### 2.5. SEM Sample Preparation of the Decellularized Tissues

Small parts of approximately 1 cm^2^ were sectioned and fixed in 4% paraformaldehyde (PFA) solution for 15 min. After washing the tissues with PBS^−^ twice, the samples were dehydrated in an ascending acetone series. After a critical point drying process (CPD 030, Bal-Tec, Untersiemau, Germany), the tissues were placed on a SEM sample holder and were coated with 2 nm of platinum (EM ACE600, Leica, Vienna, Austria). Samples were analyzed by SEM (Ultra 25, Zeiss, Jena, Germany).

### 2.6. Sources and Isolation of Primary Human Cells

All primary cells used in this study were obtained from human biopsies with approval of the local ethics committee (vote 182/10) from the University Hospital Würzburg. Experiments were conducted in compliance with the rules for investigation on human subjects, as in the Declaration of Helsinki. Endothelial cells and fibroblasts were isolated from skin biopsies by mechanical and enzymatic treatment [15]. Human mesenchymal stem cells were isolated from bone marrow aspirate (University Hospital Würzburg, Würzburg) by Biocoll (Biochrom AG, Berlin, Germany) centrifugation as described previously [15]. 

### 2.7. Re-Endothelialization of the Scaffolds

A porcine heart scaffold was cannulated through the left coronary artery by an 18 G cannula and secured with surgical sutures. A section of the tissue was cut in the area around the artery, including parts of the left and right ventricle with the coronary in the middle resulting in a length of approximately 9 cm. Following, the tissue was mounted in a bioreactor to start a 3D dynamic culture. Flow started at a rate of 5 mL/min using a 1:1 mixture of Vasculife endothelial cells medium (Cell Systems Biotechnologie Vertrieb GmbH, Troisdorf, Germany) and DMEM containing 15% FSC overnight in an incubator (37 °C, 5% CO_2_). Human microvascular endothelial (hMVEC) cells and fibroblasts (hFibs) in a 2:1 ratio were injected through the cannula with a total number of 2 × 10^8^ cells in a 1.5 mL medium. The scaffold was incubated in a static condition to allow cell attachment overnight. Next, the flow was applied at 1 mL/min for the first day and increased daily by 1 mL/min until it reached 3 mL/min. The construct was cultured for two weeks with media changed every 5 days (*n* = 9).

### 2.8. Coagulation Test for Re-Seeded Scaffold

The loss of functional vasculature due to decellularization leads to coagulation up-on reintroduction of plasma. We carried out coagulation analyses on the re-endothelialized porcine scaffold, to check the impact of seeding endothelial cells on reversing or minimizing coagulation that might occur when peripheral blood plasma (PBP) is injected. An amount of 10 mL of human blood was centrifuged at 2500 rpm for 30 min to separate the plasma. Following, 2 mL of plasma was injected into the coronary artery and incubated for 10 min to allow coagulation (this time was calculated based on a standardized curve of plasma coagulation time established already) [19]. Subsequently, an increasing flow rate was applied, and pressure was recorded until the tissue was disconnected from the bioreactor (*n* = 5). Blood samples were collected from five different donors and were used in five different experiments.

### 2.9. Seeding of Vascularized Tissue with hiPSc-Cardiac Cells

Recellularization of the vascularized scaffold was performed on a section of the septum, as this area on one hand provides easy access to the vasculature tree and on the other hand, comprises different myocardial regions with differing tissue microenvironments and cues. A diverse microenvironment is of interest, as it is known that iPSc-CC represents a heterogenic culture that can result in the derivation of different cardiac cells with differing demands on their microenvironment. Two weeks after endothelialization, the introduction of hiPSc-CC was applied as previously described and established by Guyette, J.P. [13] by injecting a co-culture of hFibs (0.7 × 10^7^ cells/mL), hMSC (0.7 × 10^7^ cells/mL), and hiPSC-CC (1.4 × 10^7^ cells/mL) at a ratio of 1:1:2 and a seeding density of 6 × 10^6^ cells/cm^2^ intramurally. Cells were injected (100 µL per injection) in different areas of the right and left ventricle with a total volume of 2 mL cell suspension. The tissue was mounted back into the bioreactor for dynamic 3D culture, applying a flow rate of 5 mL/min (*n* = 9). The following day, some constructs (*n* = 4) were switched into a static culture by mounting pieces of it on metal cell crowns (culture area 1.1 cm^2^) [20]. This allowed cells to distribute and align better by stretching the scaffold clamped in a cell crown. 

### 2.10. Preparation and Staining of the Generated Cardiac Patches

Recellularized scaffolds were removed from 3D dynamic culture and static cell crown cultures after up to 8 weeks. For the dynamic culture, tissue fixative was injected into the coronaries to ensure the fixation of internal areas and blood vessels. Small sections of 2 cm length were cut and placed in falcon tubes filled with fixative solution 4% PFA for 24 h on a shaker at 4 °C. Static cultured tissues were fixed in 4% PFA for 6–8 h on a shaker at room temperature. Tissues were paraffin-embedded and later cut into 3 μm-thick sections for slide preparation. Afterward, H&E, Trichrome, and Van Gieson staining were performed.

For immunofluorescence staining, sections were blocked with 5% donkey serum in PBS^−^ containing 0.5% Tween 20 for one hour at room temperature. Following, primary antibodies CD90 (Abcam, Cambridge, UK, ab92574, 1:200), Von Willebrand factor, (Abcam, Cambridge, UK, Ab6994, 1:100), Vimentin (Abcam, Cambridge, UK, EPR3776, 1:100), α-SMA (Sigma-Aldrich Chemie GmbH, München, Germany, 1:100 A5228) and cardiac Troponin T (cTnT), (Sigma-Aldrich Chemie GmbH, München, Germany HPA0015774, 1:1000) were applied in a humidified chamber at 4 °C for overnight. Afterward, sections were incubated with secondary antibodies for one hour at room temperature in the dark. After washing with PSB^−^-Tween-20, sections were mounted with Mowiol-DAPI (Southern Biotech, Birmingham, AL, USA). Fluorescence images were taken by the BZ-9000 BIOREVO system (KEYENCE, Neu-Isenburg, Germany) using the standard microscope software. Images stained with secondary antibodies only were used as a control to exclude unspecific binding of the secondary antibody. For quantification, CD90- and cTnT-positive cells were counted manually using DAPI counterstained images (3 images of 1 biological replicate for each time point). The number of positive cells is given as the percentage of DAPI positive nuclei counts. 

### 2.11. Assessment of Tissue Contraction

Changes in optical density (OD) of random areas, spanning the contracting tissue in the video clip were measured using either Axiovision 4.3.2 Zeiss GmbH software or Image J. The OD values were inverted and the number of peaks in the recorded clip was converted into relative beats per min according to the following formula [{(number of peaks × video rate)/total number of frames in the recording} × 60. A representative video clip is included in the Appendix A. To include the variance, we have presented the contraction data as mean ± std which was 25.61 ± 5.99 (*n* = 3). 

## 3. Results

For the generation of decellularized cardiac tissue (Figure 1A) two different protocols were applied. The perfusion with 4% SDO for 7 days followed by 24 h of 1% Triton X-100 and 7 days PBS^−^/antibiotics resulted in partial decellularization with limited cell removal even after one week (protocol 1). The resultant scaffold was clear at the outer surface area with light pinkish color remaining inside (Figure 1B). With protocol 2, a more efficient removal of cells was accomplished, already after 3 days of 1% SDS treatment, followed by 24 h of Triton X-100 and finally washing with PBS^−^/antibiotic to flush out SDS and cell debris for an additional 10 days. Just a few colored residues remained on day 4 in the heart tissue (Appendix A) and the scaffold turned white with transparent structures until the end of protocol 2. In comparison to protocol 1, protocol 2 resulted in a whitish tissue, as seen on the images of days 7 and 14 (Figure 1B). The recorded flow rate (Figure 1C) in the vasculature under constant pressure conditions indicated vanishingly small changes when applying 4% SDO (50–68 mL/min) throughout the decellularization process. In contrast, perfusion of 1% SDS changed the flow rate significantly by time during the cell removal process, starting at 60 mL/min and reaching a maximum of 160 mL/min already at day 8, and then remained constant until day 14 (Figure 1C).

On a microscopic level, the more effective cell removal with 1% SDS compared to 4% SDO could be shown by H&E staining (Figure 1D). These observations were confirmed by quantification of the DNA content (Figure 1E), which revealed more efficient DNA clearance in hearts that were decellularized with SDS compared to scaffolds generated with SDO. Importantly, the residual DNA content was significantly decreased (38.33 ± 2.01 ng/mg tissue for SDO and 5.97 ± 0.7 ng/mg tissue for SDS) and thereby in the case of protocol 2 successfully remained under the threshold of 20 ng/mg DNA required for decellularized tissue [13].

Immunohistochemistry confirmed the presence of collagen I, fibronectin, and elastin within scaffolds from both preparations (Figure 1F). This was further settled by quantitative assay for total glycosaminoglycan (Figure 1G), which revealed similar levels of total GAGs with no significant difference (6.83 ± 0.96 μg/mg tissue for SDO and 5.4 ± 1.8 μg/mg tissue for SDS) for both decellularization protocols. Compared to cadaveric heart tissue, total GAGs were reduced by 35% for the use of 4% SDO and 50% for 1% SDS. Overall, protocol 1 demonstrated adverse decellularization results concerning cell removal and chemical exposure time. Thus, all further investigations were performed with decellularized tissues obtained using protocol 2.

Previous work demonstrated the feasibility to generate vascularized cardiac tissue based on SIS [15], a widely applied scaffold material for various in vitro tissue engineering and even in vivo applications. To show the differences between decellularized SIS and cardiac tissue, the mechanical properties were investigated by a stress–strain experiment. Recorded stress–strain characteristics showed two areas with a constant slope (Figure 2A). Furthermore, the appearance varied during the measurement from a gelatinous to a fibrous morphology. As the decellularized tissue contains about 95% water, the first constant slope area describes Young’s modulus of the tissue/water construct. When exceeding the first constant slope area, more and more fluid was pressed out of the tissue. In the second constant slope area, the water content is dramatically reduced. Thereby, the tension acted directly on the ECM fibers until tissue ruptured. The graph shows clear differences between the two decellularized tissues. The Young’s modulus of the gelatinous area (Figure 2B), as well as the fibrous area (Figure 2C) is at least three times decreased in the cardiac tissue. As the different mechanical behavior of the scaffolds might be caused by a differing ECM architecture, the ultrastructure of the ECM was observed by SEM. While the SIS features a highly orientated ECM structure (Figure 2D), the cardiac tissue had a decreased degree of ECM fiber orientation (Figure 2E). Furthermore, the protein fibers were quite fluffy in the cardiac ECM, whereas the protein fibers in the SIS ECM were stuck together to some extent by small sheet-like layers, presumably presenting parts of the basal lamina. In contrast, no residues of such sheet-like structures were visible in the cardiac ECM. This led to an improved fibrous structure in the cardiac ECM and might be an aspect for improved cardiac cell interaction and tissue contraction.

Next, we aimed to restore the vascular function of the scaffold by recellularization of the vessel structures with hMVEC and hFibs. To reduce the cell number required for tissue re-endothelialization, a section of the tissue was cut in the area around the artery, including parts of the left ventricle and right ventricle with the coronary in the middle, before cell injection (Figure 3A). Analysis of the coagulation of injected plasma inside the vascular structures of cadaveric cardiac tissue showed a fluctuating pressure between 15 and 35 mmHg at flow rates between 50 and 90 mL/min (Figure 3B). When testing the coagulation on the decellularized cardiac tissue, a high-pressure range between 200–580 mmHg was recorded throughout the experiment after incubation of plasma, indicating pronounced clot formation in the vasculature tree that caused increased internal pressure. When the pressure reached 580 mmHg for a flow rate of 2 mL/min, the cannula disconnected from the tissue, as resistance was reaching maximal tolerable pressure. For the re-endothelialized cardiac tissue, recorded pressure varied between 50–130 mmHg for flow rates up to 30 mL/min, indicating a significantly reduced coagulation (Figure 3C).

As the coagulation test indicated partial re-endothelialization, the tissues were analyzed by histology. Van Gieson’s staining demonstrated cell-populated blood vessels and an unchanged collagen structure in comparison to the decellularized scaffold (Figure 4A,B). H&E staining of cardiac tissue visualized endothelialization of differently sized blood vessels in different parts of the tissue (Figure 4C,D). In addition, Trichrome staining (Figure 4E,F) marked the cellular content in the re-endothelialized scaffold. No cells were found outside the vessel structure, indicating that the hFibs might stay around re-endothelialized tubular structures. Staining of the endothelial marker von Willebrand factor (VWF) (Figure 4G), as well as alpha-smooth muscle actin for labeling of hFibs (Figure 4H) confirmed the successful re-endothelialization of the cardiac dECM vessels. Additionally, stainings demonstrate a restriction of the injected endothelial cells to the vasculature. 

After endothelialization, hiPSc-CC, hFibs, and hMSC were applied to the scaffold to generate an active, human, and vascularized cardiac construct. Immunofluorescent staining of the re-seeded scaffold after 1 week demonstrated a high frequency of CD90-positive cells (hMSC), and fewer cTnT-positive cells (hiPSc-CC) (Figure 5A). At week 3, signals for cTnT increased, while CD90 staining was less compared to week 1. At week 6, cTnT-positive cells were more frequently observed compared to weeks 1 and 3, whereas no CD90-positive cells were detected. Quantification of the fluorescent images for cTnT- and CD90-positive cells confirmed this observation (Appendix A). The ratio of cTnT-positive increased from 18% at week 1 after injection to 87% at week 6, suggesting an increasing frequency of cardiomyocytes in the scaffolds. In contrast, the amount of CD90-positive cells showed an opposed progression and declined from 82% at week 1 until no CD90 expression could be detected at week 6, indicating that the injected hMSC lose their phenotype with ongoing culture. Staining for the hFibs marker Vimentin demonstrated a low signal at week 1, which increased over time, indicating a growing number of hFibs over time (Figure 5B). 

The contraction frequency was determined by recording the contracting tissue with a video camera. Changes in optical density (OD) of random areas spanning the contracting tissue in the video clip were then measured using ImageJ software (Figure 6). 

Based on the recorded OD measurement over time, a frequency of 25.61 ± 5.99 beats/min was observed. Contraction was asynchronous at the measured area (Appendix A). 

## 4. Discussion

Our previous work demonstrated the development of a human cardiac patch using a biological collagen-based vascularized scaffold (BioVasc). The scaffold was pre-endothelialized with hMVEC, allowing a physiological blood tissue interface before introducing a co-culture of hFibs, hiPSc-CC, and hMSC. Two weeks later, the tissue exhibited physiological cardiac functions, including cardiac markers expression, physiological beating rate, drugs, and an electrical stimulation response [15]. Although the scaffold facilitated a functional vasculature, which has been reported for the first time, and long-term functionality over months, the underlying scaffold did not originate from the heart and thus does not represent the specific features of the heart dECM. SIS is derived from porcine intestine and a simple characterization of the mechanical properties of SIS and decellularized cardiac tissue revealed a significant difference between SIS and decellularized cardiac muscle. In the native state, the rigidity of the heart is mostly determined by the resident muscle cells and their contractile forces rather than by the ECM components [21]. Accordingly, the produced decellularized heart was characterized as a very soft scaffold compared to the stiffer SIS, indicating the preservation of tissue-specific ECM cues. The lower stiffness of the heart ECM is presumably due to a decreased fiber orientation compared to the SIS as well as the comprehensive preservation of elastin, which is an important mediator of tissue elasticity and thereby facilitates myo- and epicardial contractility. In addition to distinct mechanical properties, also ultrastructural differences between SIS and heart dECM were observed, which presumably arise from diverse ECM compositions. The tissue-specific substrate elasticity and ECM composition have been shown to be important orientation marks for cellular development [22]. Therefore, the creation of scaffolds, ideally from the heart tissue itself, that meet the biophysical and biochemical parameters of the original tissue, appears as an elementary factor to generate functional tissues for cell replacement approaches. 

A major criterion for the generation of heart scaffolds with biochemical and biophysical properties is the preservation of the ECM architecture and composition during scaffold production. This requires fine-tuned protocols that provide efficient cell removal with minimal ECM disruption. Previous approaches for the decellularization of whole hearts relied on a combination of mechanical, chemical (SDS, SDO, Triton X-100), and enzymatic (Trypsin) methods for decellularization, which propagated ECM protein deterioration and the loss of biochemical cues. Methe et al. [9] identified SDS to be the most effective anionic detergent in removing cell nuclei material from dense tissue, such as kidney and heart, however with an adverse elimination of growth factors and a disruption of matrix ultrastructure [13]. In accordance with that, our comparison of two decellularization protocols demonstrates that SDS is superior to SDO in terms of cell removal efficiency and thus confirming that SDS presents a suitable detergent for the decellularization of whole porcine hearts. In contrast to other protocols [13,18], efficient cell removal could be achieved with reduced exposure times and without the use of enzymatic and mechanical stimulation. The rationale for the generation of a cardiac patch is the treatment of ischemic cardiac areas. Thus, a suitable nutrient and oxygen supply must be established immediately after implantation to support cell survival. This is possible if the patch comprises a functional vascular structure that is anastomosed to the host vasculature during implantation. The presented protocol enables the preservation of vessel structures, and the coagulation assay showed a reduced clogging after a partial repopulation of the embedded vasculature. As the maximum pressures for re-endothelialized scaffolds were still higher compared to cadaveric tissue, the number of seeded endothelial cells should be increased to facilitate a better repopulation of the vessels in the future.

For the accomplishment of a functional and implantable cardiac patch, the proper maturation of the tissue and, thereby long-term survival of the graft, needs to be enabled. In the study presented here, we observed the reduction and vanishing of CD90 with a simultaneous increase of cTnT. A factor that might facilitate the observed differentiation of stem cells into cells with a cardiac character is the dECM scaffold. Several studies emphasized the effect of the ECM on stem cell differentiation [22,23]. For instance, Goetzke et al. [23] showed that hiPSc differentiation within 3D fibrin hydrogels promotes neural development and results in altered DNA methylation profiles within genes for cardiovascular and neuronal development, indicating that the ECM environment contributes to stem cell lineage decisions. Other studies reported that individual properties of the ECM, such as matrix elasticity, nanotopography, and certain ECM components, can direct the lineage decision of stem cells [22,24,25]. To which extent the here generated cardiac dECM contributes to a cardiac cell fate of hMSC, and hiPSc remains an open point and presents an interesting starting point for follow-up studies. Besides the matrix influence, the chemical composition of the culture medium was also shown to promote differentiation of hMSC into cardiac myoblasts [26]. Moreover, co-culture of hMSC with cardiac cells [27] or stimulation with demethylating agent 5-azacytidine [28] enhances the differentiation of hMSC into cardiac myoblasts, as shown in several studies. Similarly, this was observed in our previous study [15], where CD90-positive cells start decreasing post-seeding on the SIS scaffold. Thus, it is conceivable that the culture conditions used in our study promote the cardiomyoblasts differentiation of hMSC.

The resulted tissue started beating in an unsynchronized rhythm within two days of culture. In our previous study [15], we observed that the SIS cardiac patch contraction was synchronized at a later point when cells were fully maturated. Therefore, asynchronous contraction is reasoned by a certain cellular immaturity and is expected for synchronization on a further advanced cardiac maturation level. Moreover, histological staining showed an alignment of cells in dense tissue architecture and agrees with previous studies [27,28,29]. Furthermore, hiPSc-CC in combination with hFibs and hMSC applied in decellularized cardiac tissues resulted in cell-matrix coupling with a spontaneous rhythmic contraction of about 25.61 beats/min. Fibroblasts and mesenchymal stem cells have a crucial role in regulating myocardial contraction response, electrical cardiac conduction, and enhancing ventricular function. Their transdifferentiation into cardiac-like cells, kinases paracrine release, and/or a long-term Wnt signaling pathway inhibitory effect promotes antiapoptotic effects [29,30]. Hence, we assume that the co-culturing of both fibroblasts and hMSC along with hiPSc-CC was important for optimizing the active cardiac function of the recellularized cardiac scaffold.

The final mentioned criterion for the development of cardiac patches is the transferability for clinical use. Thereby, a personalized and autologous approach is preferred, to minimize rejection reactions after implantation. The resultant decellularized porcine heart matrices contained less DNA than the currently accepted standard for implants [13] and could therefore be more compatible with clinical transplantation. A bioengineered myocardium using a patient’s own (autologous) stem cells, like hiPSc, provides a solution to overcome many transplantations hurdles. Most studies are based on the injection of contracting cardiac cells, originating from embryonic stem cells (ECS) and biomaterials like matrigel [31,32]. However, these approaches fail to fulfill all requirements, such as biodegradability in host tissue, consistent mechanical properties, and appropriate scaffold architecture [33,34].

## 5. Conclusions

In conclusion, the present study reports a simple tailored decellularization protocol for porcine hearts using only ionic SDS + Triton-X-100 perfusion, without any mechanical stimulation, and less detergent exposure time. The resulted scaffolds were cell-free, with major extracellular matrix components preserved, such as glycosaminoglycans, collagens, and elastin. By optimizing the decellularization steps through reducing the exposure time of SDS, it is possible to mitigate negative effects on essential ECM proteins and biochemical cues. As a result, the produced scaffold exhibits an ECM architecture with a homogenous distribution of collagen, fibronectin, and elastin. Moreover, the ECM supported the differentiation of stem cells into cardiac-like cell types and a spontaneously beating tissue was obtained. The presented approach to measuring the contraction frequency using image analysis renders an alternative to classical multi-electrode measurements. The engineered cardiac tissue has the advantage of pre-repopulation of the vasculature with CD31-positive cells, which enhances the probability of cell survival within the construct when implanted. Using the novel clogging assay, the functionality of the partially reseeded vasculature was demonstrated. By applying dynamic culture with perfusion of the vasculature, long-term culture was possible. The simple and time-efficient protocol, the well-preserved ECM, the successful generation of human cardiac tissue that was cultured for weeks as well as the functional vasculature with central access for connecting the tissue to a host’s circulatory system will support the survival of the cardiac tissue when considered for clinical applications. Moreover, the patch renders a tool for drug development. Nevertheless, further physiological characterization of the patch, testing its response drugs, and animal tests are needed.

## Figures and Tables

**Figure 1 bioengineering-09-00147-f001:**
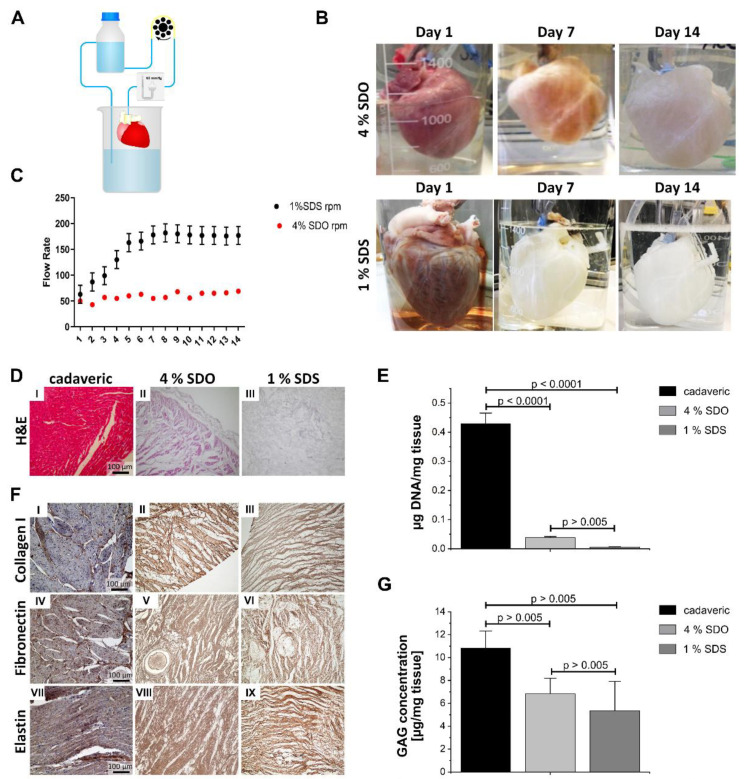
Decellularization of porcine hearts. (**A**) Schematic setup of the decellularization setup. (**B**) Change in appearance of the porcine hearts during the decellularization under constant pressure (60 mmHg) using two different detergent protocols: (1.) SDO 4% for 7 days, followed by 1% Triton X-100 for 24 h, then 7 days rinsed with PBS^−^. (2.) 1% SDS for 3 days, followed by 1% Triton X-100 for 24 h, then excessive rinsing with PBS^−^ for 10 days (representative images of *n* = 20). (**C**) Recorded pump rate during the constant pressure decellularization process (*n* = 4). (**D**) H&E staining of cadaveric (I) and decellularized cardiac tissue. 1% SDS (III) showed more cell removal than the treatment with 4% SDO (II) (representative images of *n* = 5). (**E**) Measured DNA concentration in cadaveric (0.43 µg/mg) and decellularized tissue (SDO: 0.038 µg/mg, SDS: 0.006 µg/mg). Not significant *p*-value > 0.005; (*n* = 3). (**F**) Representative stainings of the decellularized tissue for components of the extracellular matrix indicates the preservation of collagen I (II and III), fibronectin (V and VI), and elastin (VIII and IX) in both SDS and SDO treated scaffolds in comparison to cadaveric heart tissue (I, IV and VII) (*n* = 5 for decellularized scaffolds, *n* = 2 for cadaveric heart tissue). (**G**) Total GAGs concentration in cadaveric (10.82 µg/mg), decellularized scaffold by 4% SDO (6.83 µg/mL), decellularized scaffold by 1% SDS (5.36 µg/mL) (*n* = 3). Not significant *p*-value > 0.005. Tissue analysis from different preparations was obtained on day 14.

**Figure 2 bioengineering-09-00147-f002:**
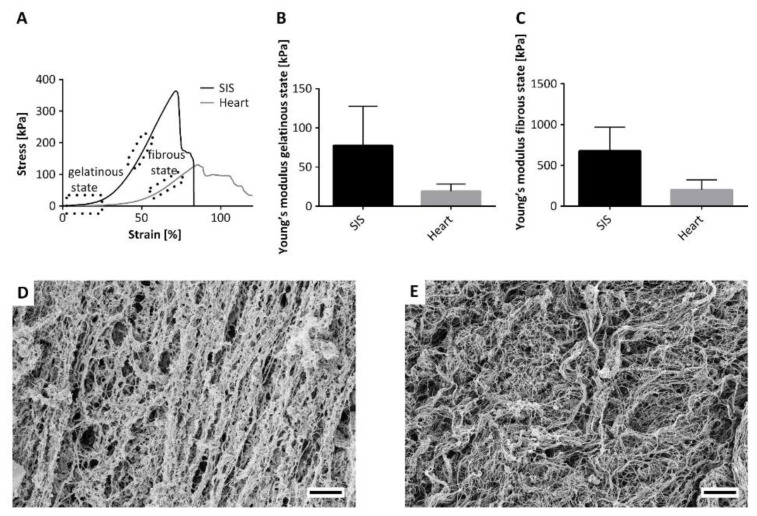
Mechanical and structural comparison of decellularized tissue from the small intestine (SIS) and the heart. (**A**) Stress–strain diagram of the tissues demonstrates two main regions in the mechanical behavior. At the beginning, the high-water content causes a gelatinous tissue condition. The following tensile stress reduced the water content. This leads to an increasing fibrous tissue behavior. (**B**,**C**) represent the Youngs‘ modulus determined from the slope of the gelatinous and fibrous sector, respectively. (**D**,**E**) Depiction of the fibrous structure by SEM of the SIS and the heart, respectively. Scale bar equals 5 µm.

**Figure 3 bioengineering-09-00147-f003:**
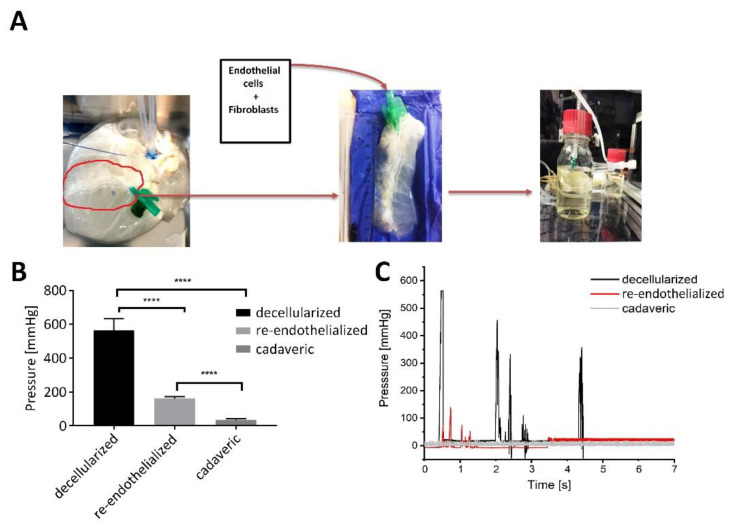
Re-endothelialization of the decellularized porcine scaffold. (**A**) Steps of porcine scaffold re-endothelialization process: preparation of tissue sections of approximately 9 cm in length around the left coronary artery, including part of the right and left ventricles with a width of 2 cm from each side. hFibs and hMVEC (ratio 1:2) were injected into the coronary artery. Following, the seeded construct was mounted in a bioreactor for 3D culture. The bioreactor facilitated the perfusion of the vasculature with a dynamic pressure of 120/80 mmHg (*n* = 6). (**B**) Coagulation assay to investigate the functionality of the endothelial layer. PBP was injected in the vasculature of cadaveric, decellularized, and in re-endothelialized tissue. Graph showing maximum pressures recorded in decellularized, re-endothelialized scaffolds and in the cadaveric heart during the coagulation assay when the flow rate was stepwise increased up to maximum 2 mL/min (cell-free decellularized tissue), 30 mL/min (re-endothelialized tissue), and 90 mL/min (cadaveric tissue) until the tissue disconnected from the bioreactor. **** *p*-value < 0.0001 (*n* = 5). (**C**) Graph showing changes in pressure throughout the experiment, in decellularized, re-endothelialized scaffolds, and in the cadaveric heart (representative of *n* = 5).

**Figure 4 bioengineering-09-00147-f004:**
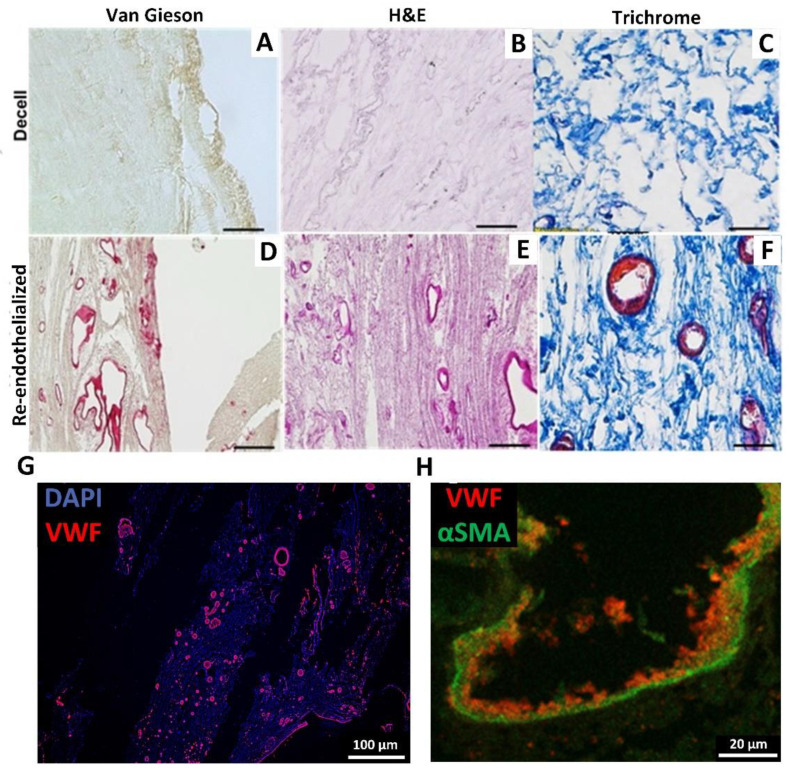
Immunohistochemical staining to characterize the re-endothelialized scaffold. (**A**–**F**) Van Gieson staining for collagen fibers in both decellularized (**A**) and re-seeded scaffolds (**B**). H&E staining of decellularized (**C**) and re-seeded scaffolds (**D**). Trichrome staining of decellularized (**E**) and re-seeded (**F**) scaffolds. Scale bars depict 100 µm. (representative images of *n* = 6). (**G**) + (**H**) Immunofluorescent staining of the scaffold after injection of hMVEC and hFibs for the endothelial cell marker VWF (**G**,**H**) (red) and the fibroblast marker αSMA (**H**) (green). Analyses were performed two weeks post-re-endothelialization.

**Figure 5 bioengineering-09-00147-f005:**
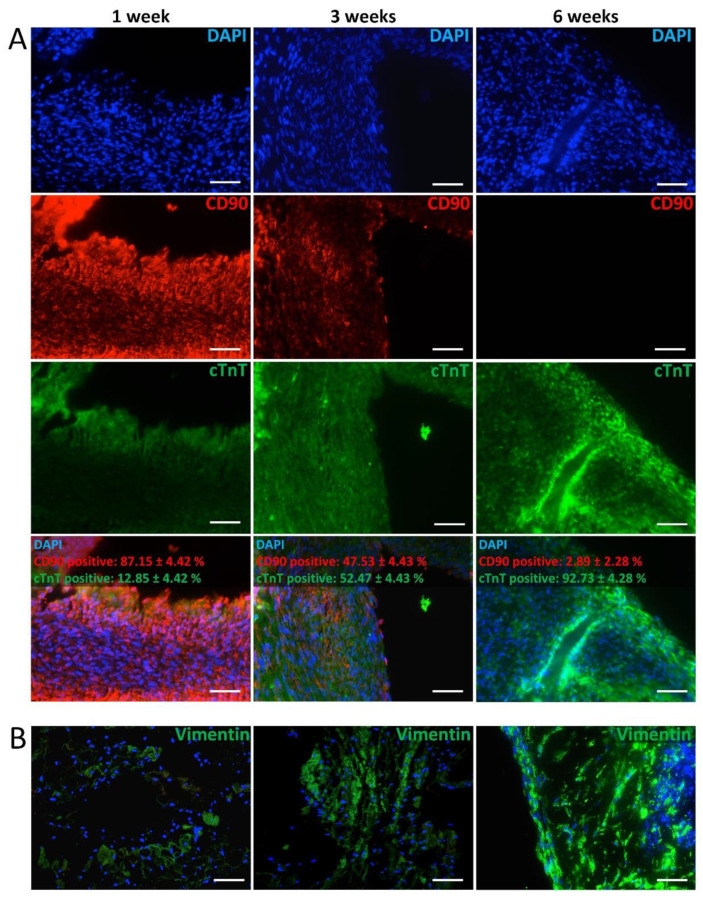
Characterization of the time-dependent cell type distribution. (**A**) Immunofluorescent staining of re-seeded cardiac tissues for CD90 (hMSC marker, red) and Troponin T (cardiac marker, green) along a period of 6 weeks culture. Sections were counterstained with DAPI. A quantification of marker-positive cells is provided in Appendix A. (**B**) Immunofluorescent stainings of sections of the re-seeded heart for Vimentin (hFibs marker, green) counterstained with DAPI. Scale bar depicts 100 µm. (representative images of *n* = 5).

**Figure 6 bioengineering-09-00147-f006:**
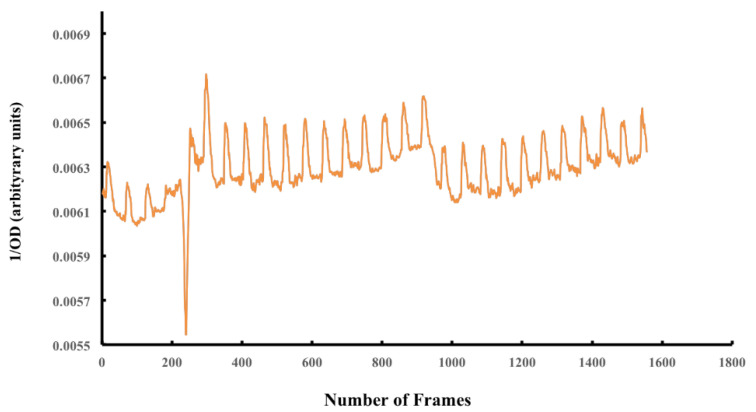
Determination of contraction frequencies of the recellularized cardiac tissues. Analysis of contraction rate for the generated tissue after day 15 of culture (see Appendix A) using ImageJ software according to measured pixel intensity in a frame range. Mean beating frequency was determined as 25.61 ± 5.99 beats/min (*n* = 3).

## Data Availability

All data is available in the paper.

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
