# Peer review of "Decellularization of Full Heart—Optimizing the Classical Sodium-Dodecyl-Sulfate-Based Decellularization Protocol"

_bioengineering, 2022, doi:10.3390/bioengineering9040147_

Round 1
Reviewer 1 Report
A lot of improvement, but there are still some important points to be added/discussed:
Material and methods:
Line 107: decellularization of the heart – please find a better to read easily and quickly that 2 different protocols are tested (i.e.: bold, or new line for each protocol etc..)
Line 123: please provide more details about gamma sterilization (machine, dose, protocol)
Line 193: “increasing rate of was applied ” seems like word is missing; “until the tissue WERE disconnected”.
Line 197: its unclear what is “reseeded” – did you seed scaffolds twice? Or do you mean recellularization? If its recellularization then you this correct denomination and/or “seeded”.
Results
Lines 298-301 and 304-307: this is discussion
Lines 328-336: this is method
What is the author’s definition of vascularized and re-vascularized? If it’s re-endothelialization then it’s not correct, as revascularization stands for an active process of new vasculature ingrowth into matrices and reconnexion to the recipient main stream.
Figure 3: replace “vascularised” by “vascularized”.
Discussion
Line 455 : then conclusion is that SDS-based protocol is the best – what is then the novelty as you stated to have only slightly modificated Ott’s protocol? What exactly is the decellularization protocol modifications that are essential in the author’s approach and justify a significant step forward? Especially for clinical application as SDS is difficult to remove and more toxic, so not the best necessarily for clinical use. If it’s comparing Ott’s protocol, even a bit modified, to another detergent-based protocol, it has been published dozen of times. This point is critical for both the reviewer and a reader to establish a true novelty in this paper. Or then the title should be more like “optimizing the classical SDS-based decellularization protocol”. Then develop more your findings, make clear and detail the improvements in the protocol as well in the discussion. And discuss the link with clinical translation and what should be the next step.
Author Response
Reviewer 1
A lot of improvement, but there are still some important points to be added/discussed:
Material and methods:
Comment 1: Line 107: decellularization of the heart – please find a better to read easily and quickly that 2 different protocols are tested (i.e.: bold, or new line for each protocol etc..)
Response to comment 1:
We thank the reviewer for this comment. We addressed the request and clearly separated and named the two protocols:
“… under constant pressure of 60 mmHg for 4 - 5 hours. Following, two protocols were applied for decellularization:
Protocol 1: 4 % SDO (Sigma-Aldrich, Munich, Germany) in MilliQ water was applied for seven days of decellularization, followed by 3 hours of MilliQ water and one day with 1 % Triton X-100 in milliQ water afterwards. Final rinsing with PBS-/antibiotics solution was preformed until day 14.
Protocol 2: 1 % SDS (Carl Roth GmbH, Karlsruhe, Germany) in MilliQ water was used to decellularize the heart for 3 days, followed by 3 hours of MilliQ water rinsing and one day of 1 % Triton X 100 (Sigma-Aldrich, Munich, Germany) in MilliQ water. At day 5, PBS-/antibiotics solution was started until day 14. In both protocols, gamma sterilization was conducted 14 days after organ explantation. “
Comment 2: Line 123: please provide more details about gamma sterilization (machine, dose, protocol)
Response to comment 2:
We thank the reviewer for this comment. We added the sterilization details in the methods:
“In both protocols, gamma sterilization was conducted 14 days after organ explantation at a dosage of >25 kG using the sterilization service provided by BBF steriXpert (Kernen-Rommelshausen, Germany) and sterilized samples stored in PBS- at 4 °C.”
Comment 3: Line 193: “increasing rate of was applied ” seems like word is missing; “until the tissue WERE disconnected”.
Response to comment 3:
We thank the reviewer for finding the missing word. We adjusted the sentence as follows: “Subsequently, an increasing flow rate was applied and pressure recorded until the tissue was disconnected from the bioreactor (n = 5).”
Reviewer 2 Report
I think the authors answered to the comments properly and the purpose and the merits are clear. So, I recommend to publish this manuscript after minor revision described below.
Please show the DNA and GAG contents as ug/tissue weight or ng/tissue weight. Do not show the concentration. Concentration can be easily changed by the dilution. More importantly, descriptions of total tissue weight and extraction solution volume were lacked. The readers cannot calculate the contents of DNA and GAG in decellularized heart. So, I recommend to show the DNA and GAG contents as content weight per tissue weight.
Author Response
Reviewer 2
I think the authors answered to the comments properly and the purpose and the merits are clear. So, I recommend to publish this manuscript after minor revision described below.
Comment 1: Please show the DNA and GAG contents as ug/tissue weight or ng/tissue weight. Do not show the concentration. Concentration can be easily changed by the dilution. More importantly, descriptions of total tissue weight and extraction solution volume were lacked. The readers cannot calculate the contents of DNA and GAG in decellularized heart. So, I recommend to show the DNA and GAG contents as content weight per tissue weight.
Response to comment 1:
We thank the Reviewer for this comment. According to the reviewer’s suggestion, we have recalculated GAG and DNA values and now display the concentration as μg/mg tissue or ng/mg tissue. In this regard, we updated Fig. 1 E and G and the legend of Fig. 1. We further integrated the new values for GAG and DNA in the result section and updated the methods part.
Result part: “Importantly, for both protocols the residual DNA content was significantly decreased (38.33 ± 2.01 ng/mg tissue for SDO and 5.97 ± 0.7 ng/mg tissue for SDS) and thereby successfully remained under the threshold of 20 ng/mg DNA required for decellularized tissue [13].
[…]
This was further settled by quantitative assay for total glycosaminoglycan (Fig. 1G), which revealed similar levels of total GAGs with no significant difference (6.83 ± 0.96 μg/mg tissue for SDO and 5.4 ± 1.8 μg/mg tissue for SDS) for both decellularization protocols. Compared to cadaveric heart tissue, total GAGs were reduced by 35 % for the use of 4 % SDO and 50 % for 1 % SDS.”
Methods part: “dsDNA was extracted from weighed wet tissue pieces using the DNeasy® Blood & Tissue Extraction Kit (Qiagen, Germany) according to the manufacturer’s guidelines. Sample elution was carried out with a volume 100 μl. DNA content was quantified using a NanoQuant Plate™ in combination with an Infinite® 200 PRO Plate Reader (TECAN, Switzerland). DNA from Native tissue was measured at a 1:5 dilution (n = 3).”
This manuscript is a resubmission of an earlier submission. The following is a list of the peer review reports and author responses from that submission.
Round 1
Reviewer 1 Report
The revision was fast and very limited and doesn't support acceptance. I suggest rejection, comparison of such 2 decellularization protocols for heart tissue, either as a patch or in toto critically lacks novelty. The other aspects of scaffold characterization as described partially should be more developed : it will make a very interesting paper then.
Reviewer 2 Report
While the authors are correct that decellularized extracellular matrices are used in clinical settings, the assumption that SIS is the only clinically available material is incorrect. There are, in fact, several decellularized myocardial materials as well, which would serve as a more robust control. At this time, the authors have not adequately addressed my (or another Reviewer's) previous comments.
Reviewer 3 Report
The authors compared SDS- or SDO-based decellularization of heart with perfusion. And they found that SDS-based decellularization could remove cells from the heart tissues with ECM proteins. Also, they reported that the cells can form neo vascular tissues. Overall, the methods compared in this manuscript have been already compared by other reports and also it is known that the cells can form neo vascular tissues. I cannot understand the originality of this manuscript. Moreover, the manuscript did not be finished for the revision. So, I think this manuscript should be rejected. Specific comments are below.
- the authors compared just 2 protocols with fixed concentrations of SDS and SDO. How did the authors claim that the decellularization method was optimized? There are several reports to compare SDS and SDO (Booth et al., J Heart Valve Dis, 2002, 11, 457-462, Remlinger et al., J Vis Exp, 2012, 6, e50059, Methe et al., Biores Open Access, 2014, 3, 327-338). Remlinger et al. reported that SDO can remove the cells from the heart tissues after protease treatment. It might be possible that this method is suitable for heart decellularization. Moreover, Methe et al. can successfully remove the cells from heart tissue with a perfusion method. Why did the authors ignore these papers?
- Why did the authors compare between their decellularized heart and SIS?
- The problems in heart decellularization technique are not clear. The authors mentioned that maintaining the 3D structure under room temperature without overlapping was challenging. How did the authors solve this problem by comparing 2 protocols? The authors should describe the problems to be solve in this study clearly. Also, the authors should describe the purpose more clearly.
- The culture in the SDS-based decellularized heart has been reported in previous reports. (Carvalho et al., J Tissue Sci Eng, 2012, Suppl 11, 002, Ott et al., Nat Med., 2008, 14 , 213-221). It is easily speculated that the cells can be cultured in the scaffolds and form new vascular tissues.
- Please complete to describe Introduction part.
- There are several typos.